rsob.royalsocietypublishing.org

**Subject Area:**
developmental biology/cellular biology/genetics

RTK, Torso, *Drosophila*, trunk, Torso-like

**Author for correspondence:**
Jordi Casanova
e-mail: jordi.casanova@irbbarcelona.org

†Present address: MRC London Institute of Medical Sciences, Du Cane Road, London W12 0NN, UK.

# The trigger (and the restriction) of Torso RTK activation

Alessandro Mineo[1,2,†], Marc Furriols[1,2] and Jordi Casanova[1,2]

[1]Institut de Biologia Molecular de Barcelona (CSIC), C/Baldiri Reixac 10, 08028 Barcelona, Catalonia, Spain
[2]Institut de Recerca Biomèdica de Barcelona, (IRB Barcelona), The Barcelona Institute of Science and Technology (BIST), C/Baldiri Reixac 10, 08028 Barcelona, Catalonia, Spain

(iD) AM, 0000-0002-7047-1313; JC, 0000-0001-6121-8589

The Torso pathway is an ideal model of receptor tyrosine kinase systems, in particular when addressing questions such as how receptor activity is turned on and, equally important, how it is restricted, how different outcomes can be generated from a single signal, and the extent to which gene regulation by signalling pathways relies on the relief of transcriptional repression. In this regard, we considered it pertinent to single out the fundamental notions learned from the Torso pathway beyond the specificities of this system (Furriols and Casanova 2003 *EMBO J.* **22**, 1947–1952. (doi:10.1093/emboj/cdg224)). Since then, the Torso system has gained relevance and its implications beyond its original involvement in morphogenesis and into many disciplines such as oncogenesis, hormone control and neurobiology are now acknowledged. Thus, we believe that it is timely to highlight additional notions supported by new findings and to draw attention to future prospects. Given the late development of research in the field, we wish to devote this review to the events leading to the activation of the Torso receptor, the main focus of our most recent work.

## 1. Introduction

The ability of cells to communicate and exchange information is a basic requirement for the development and homeostasis of organisms. This property is sustained mainly by transduction pathways that allow cells to respond to extracellular signals. One of the most common of such mechanisms involves the presence of receptor tyrosine kinases (RTKs) at the cell membrane. RTKs are eventually activated by extracellular ligands and transduce this signal by a well-conserved pathway of intracellular molecules, including the Ras/Raf/MAPK cascade, which finally elicits a range of cell responses in terms of cytoskeletal changes and/or gene activation (for a review, see [1]).

The *Drosophila* Torso pathway is an ideal model of RTK systems, in particular when addressing questions such as how receptor activity is turned on and, equally important, how it is restricted, how different outcomes can be generated from a single signal, and the extent to which gene regulation by signalling pathways relies on the relief of transcriptional repression. In this regard, we considered it pertinent to single out the fundamental notions learned from the Torso pathway beyond the specificities of this system [2]. Since then, the Torso system has gained relevance and its implications beyond its original involvement in morphogenesis and into many disciplines such as oncogenesis, hormone control and neurobiology are now acknowledged. Thus, we believe that it is timely to highlight additional notions supported by new findings and to draw attention to future prospects. Given the late development of research in the field and because a recent article reviewed the Torso downstream processes [3], we wish to devote this review to the events leading to the activation of the Torso receptor, the main focus of our most recent work.

rsob.royalsocietypublishing.org  Open Biol. **8**: 180180

## 2. A two-part mechanism to transfer information between tissues

Torso was initially identified as the receptor of a signalling pathway that transduces positional information from the ovary into the embryo [4]. In particular, a distinct group of ovarian follicle cells at both poles of the oocyte are ultimately responsible for the activation of the Torso receptor in the most anterior and posterior parts of the embryo [5]. Once activated, the Torso receptor triggers the genetic programme that specifies the development of the two embryonic termini. For that reason, the Torso transduction pathway is also known as the terminal system. The Torso receptor accumulates all over the early embryo and becomes activated only at its poles [6–8]. But, how does the receptor become activated and how is this activation limited to the poles?

This process is achieved by two separate elements. On the one hand, Trunk is singled out as the ligand for Torso [9] and thus as the molecule that binds and activates the receptor. On the other hand, many results clearly establish that another molecule, Torso-like (Tsl), is responsible for having the receptor exclusively activated at the poles and only at the poles [10,11], probably by acting upon Trunk. Below we first discuss current knowledge about these two elements and then address the hypotheses put forward to explain how these two elements are integrated into a common mechanism.

## 3. Generating an active ligand by cleavage

Although a formal proof that establishes Trunk as a ligand for the Torso receptor is still lacking, all experimental results strongly support this notion. In this regard, the following has been established: (i) Trunk is a secreted protein that harbours the cysteine-knot motif present in a family of ligands and growth factors [9], and (ii) the Trunk protein has a series of cleavage sites, and artificially truncated C-terminal forms of Trunk are sufficient to elicit the outcomes of the Torso receptor independently of any of the other elements involved in the physiological activation of the receptor [12]. Indeed, while a potential proteolytic site in Trunk was shown to be required for its function some years ago [9], only recently have the cleaved forms of Trunk been identified *in vivo* [13].

Another set of features of Trunk account for an additional property of the terminal system. In particular, *trunk* RNA accumulates at the oocyte and in the early embryo, and the Trunk protein bears a signal peptide [9]. Therefore, activation of the Torso pathway results from autocrine signalling in which the same cell that accumulates the Torso receptor secretes its ligand. While it has not yet been possible to directly visualize Trunk secretion, a tagged version of an N-terminal segment of Trunk has been found to accumulate extracellularly [14]. This result is consistent with the autocrine notion of Torso activation. However, the question remains as to how an autocrine mechanism is consistent with the role of the ovarian follicle cells in Torso receptor activation. This question is especially intriguing considering that the Torso receptor is activated in the embryo, which is not in contact with ovarian follicle cells. To answer this question, we will now focus on the other element of the two-part mechanism, namely the Tsl protein.

## 4. Temporal gap and spatial restriction in receptor activation

Tsl is the protein that allows positional information to be transferred from the ovaries to later activation of the Torso receptor in the early embryo. Of note, the same molecule that bridges the temporal gap between the ovary and embryo is also the one that restricts the activation of the Torso receptor to only the embryonic poles. Tsl is synthesized by the ovarian follicle cells around the two poles of the oocyte. In the ovary, the oocyte is surrounded by a single layer of follicle cells but only a subset of these, some at the most anterior edge and some at the most posterior edge, actually transcribe the *tsl* gene and synthesize the Tsl protein. Forced unrestricted *tsl* expression in all the follicle cells surrounding the oocyte later triggers the unrestricted activation of the Torso receptor throughout the embryo, which gives rise to an expansion of terminal structures at the expense of all the other segments [10,11,15]. Thus, restricted *tsl* expression is the ultimate event leading to the spatial restriction of the Torso receptor.

Once synthesized in the follicle cells, Tsl is secreted, and accumulates at the poles of the assembling eggshell, a specialized extracellular matrix (ECM) that receives contributions from both the ovary and the oocyte [15,16]. Tsl remains at the eggshell until the embryo begins to develop, at which point it translocates into the embryonic plasma membrane. As Tsl is present only at the poles of the eggshell, once translocated, this protein accumulates exclusively at the membrane at the embryonic poles [17]. This is the stage when the Torso receptor is activated [18].

## 5. Taking advantage of ECM proteins for receptor activation

Anchoring to and release from the eggshell are therefore crucial events in the timely and spatial setting of Torso receptor activation. This is made possible by Tsl accumulation taking advantage of a set of proteins (named Nasrat, Polehole and Closca) that are secreted from the oocyte in mid-oogenesis and are mutually required for their incorporation into the eggshell. In the absence of this set of proteins, Tsl fails to accumulate at the eggshell, which thus should preclude Torso activation [16,19]. However, activation of the Torso receptor cannot be assessed in this condition because, in the absence of Nasrat, Polehole and Closca, the eggshell defects prompt eggs to collapse, thereby preventing further development [19,20]. Interestingly, some mutant variants of the Nasrat, Polehole and Closca proteins ensure eggshell formation but alter Tsl accumulation and function. In these cases, embryo development can proceed but Torso receptor activation is impaired [15,16,19,20]. The interaction between Nasrat, Polehole and Closca and Tsl is relatively specific because Tsl continues to accumulate at the eggshell in the absence of other eggshell proteins [21]. Hence, the terminal signalling pathway appears to have co-opted already existing ECM proteins to anchor Tsl, thus allowing the temporal bridge and spatial restriction mechanisms of Torso receptor activation.

rsob.royalsocietypublishing.org   Open Biol. **8**: 180180

# 6. Closing the gap: delivering a local signal or locally processing a uniform signal?

Following the steps described above, the two elements involved in the two-part mechanism get into position at the time of Torso receptor activation: Tsl accumulated at the poles of the embryonic membrane and Trunk secreted from the embryo. As mentioned earlier, all the data point to a cleaved form of Trunk acting as the Torso receptor ligand. However, there is no clear indication of the role of Tsl. Tsl harbours a membrane-attack complex/perforin domain (MACPF) [22], a domain present in proteins involved in pore formation at the plasma membrane (for a review, see [23]), which is consistent with the accumulation of Tsl at the membrane. In addition, Tsl accumulation at the membrane does not reflect its binding to the Torso receptor, as it also accumulates in embryos devoid of Torso receptor [17]. Both features are at odds with Tsl being another ligand for the Torso receptor. Instead, the focus shifted into how Tsl might contribute to enable Trunk to bind and activate the Torso receptor (figure 1).

The above-mentioned molecular features of Trunk and Tsl led to the proposal that Tsl might participate in restricting where or how much of the cleaved form of Trunk is generated or in facilitating the ability of Trunk to interact with Torso [9]. Furthermore, the observation that a cleaved form of the Trunk protein was able to activate the Torso receptor in the absence of Tsl protein [12] prompted the notion that Tsl might act as a membrane-bound protein necessary to nucleate a putative protease complex to cleave the Trunk protein, which would then behave as the ligand for the Torso receptor [2]. In the absence of any identified protease of the putative complex and of any indication of the mode of action of Tsl, this model was raised by analogy to the mechanism leading to the activation of Toll by its ligand Spätzle [24].

Indeed, while recent experiments have provided clear proof of Trunk cleavage, they have failed to show any of such proteolytic events to be Tsl-dependent [13]. Interestingly, however, Tsl has been reported to enhance the extracellular accumulation of a tagged version of an N-terminal segment of Trunk at the poles of the embryo [14]. Although this N-terminal segment does not match the active Trunk fragment, this observation has prompted the hypothesis that Tsl serves to enhance the extracellular accumulation of the active Trunk ligand at the embryo termini [14]. In particular, given that Tsl is related to membrane pore proteins [22], it has been proposed that Tsl might play a role in the secretion of Trunk from inside the embryo specifically at the poles, via a pore-forming or a membrane-damaging mechanism. Therefore, restricted accumulation of Tsl at the poles would lead to enhanced polar secretion of Trunk and thus to Trunk extracellular polar accumulation. In other settings, Tsl might influence the secretion of other growth factors [14,25].

The above scenario would replace that of the general secretion of Trunk followed by its local cleavage at the poles. But within this new scenario, how does Trunk cleavage fit into the mechanism of Torso receptor activation? In answer to this question, results from the same authors further suggest that Trunk is uniformly cleaved intracellularly by furin proteases to generate the active ligand and that this

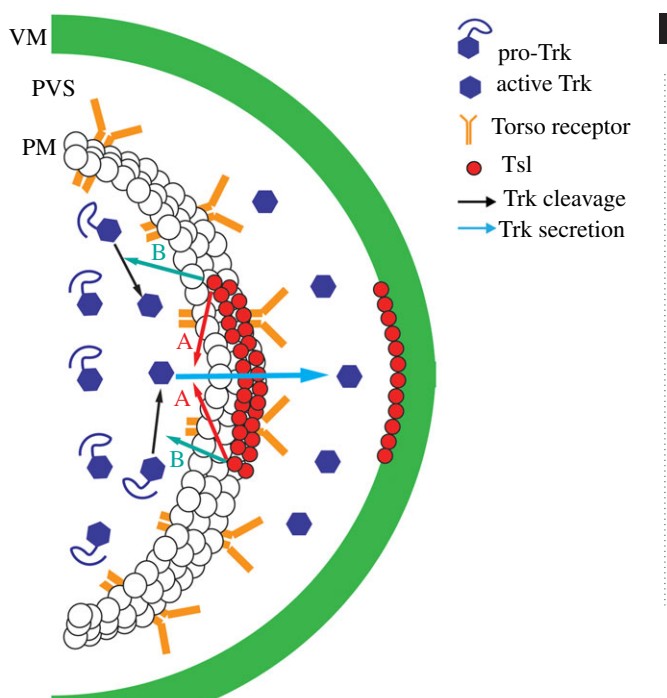

**Figure 1.** Schematic of the two main roles proposed for Tsl in Torso activation. Tsl might be involved either in Trunk (Trk) activation (B), in secretion of an activated Trunk (A) or in both as Trunk secretion might be linked to its activation. We want to note that this schematic does not include all the possibilities, for example whether Trunk activation only involves cleavage or whether Trunk cleavage/activation occurs intracellularly or extracellularly (see box 1). VM, vitelline membrane; PVS, perivitelline space; PM, plasma membrane.

active form is secreted exclusively (or almost exclusively) at the poles [14]. Thus, there is restricted secretion rather than restricted cleavage. This is an appealing model that might be supported by future experiments.

Regardless of whether Tsl is required for Trunk cleavage or Trunk secretion, does the MACPF domain of Tsl imply that this protein enables traffic between the inside of the embryo and its surrounding extracellular environment? Indeed, as previous studies addressing whether other perforins might substitute for Tsl in Torso signalling have so far proven unsuccessful (T. Johnson 2018, personal communication), we recently took an alternative approach and examined whether Tsl function might be substituted by mechanically induced holes. In fact, this was found to be the case as *tsl* mutant embryos, either pierced at the posterior pole at the blastoderm stage with a sharpened capillary similar to the ones used to generate transgenic flies, or subjected to electroporation, developed some of the terminal structures that are absent in *tsl* mutants. The development of terminal structures by mechanical induction of holes, although at low penetrance, mimics Tsl function as it requires the presence of the Torso receptor and of Trunk protein [26]. Thus, these results support the proposed notion that Tsl is involved in a mechanism that enables exchange between the interior of the blastoderm and its surrounding extracellular environment and that this exchange somehow allows the Trunk protein to activate the Torso receptor.

However, very recently, all the previous models have been called into question upon the observation that Tsl protein might activate the Torso receptor in the absence of the Trunk ligand [27]. While, on the one hand, these

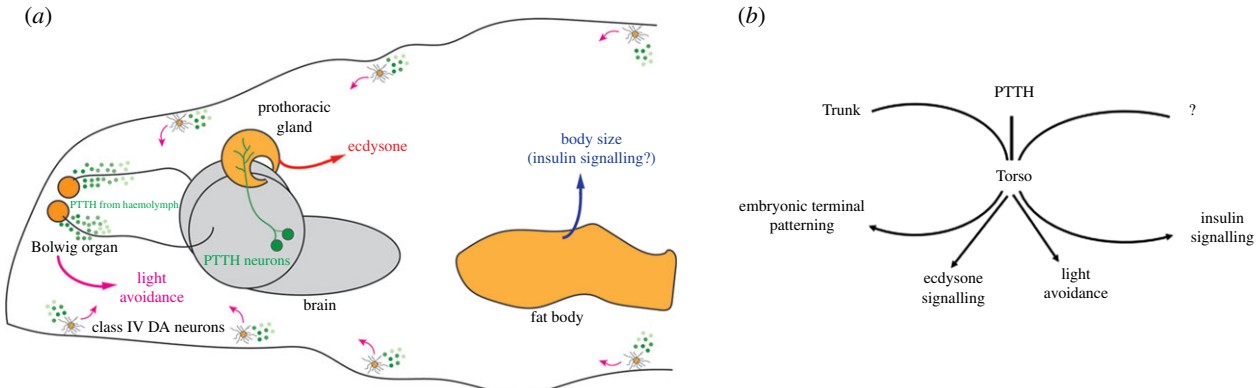

**Figure 2.** Torso function in larval tissues. (a) During larval development Torso function (in orange) is required in the prothoracic gland (where it regulates ecdysone synthesis, in red), in the fat body (where it regulates body size, probably through the regulation of the insulin/TOR signalling, in blue) and in two light sensors, the peripheral class IV dendritic arborization neurons (class IV DA neurons) and the Bolwig organ (where it regulates light avoidance, in pink). In the prothoracic gland, the class IV DA neurons and the Bolwig organ Torso is activated by PTTH (in green). PTTH-producing neurons directly innervate the prothoracic gland, whereas PTTH reaches class IV DA neurons and the Bolwig organ through the haemolymph. The ligand of Torso in the fat body has not been addressed. (b) Schematic of the Torso activating ligands in the different settings.

rsob.royalsocietypublishing.org    Open Biol. **8**: 180180

observations clearly open new perspectives, on the other hand, they have been obtained in transformed cell cultures in non-physiological high concentrations of the Torso receptor, and the authors acknowledge that these novel observations may be limited to conditions in cultured cells that do not reflect the situation in the embryo [27]. Consistent with this possibility, in the embryo, Tsl does not activate the Torso receptor in the absence of Trunk, even when Tsl is expressed at very high levels [26].

# 7. Torso activation in other settings

Recent work has provided key insights into the Torso pathway, revealing it to be active in additional signalling processes in several larval *Drosophila* tissues (figure 2). In particular, the Torso pathway is active/present in the prothoracic gland, in the fat body, in the Bolwig organ and in the peripheral class IV dendritic arborization neurons [28–30].

Activation of the Torso receptor in the prothoracic gland is triggered by a signal coming from adjacent neurons. Once activated, the Torso pathway then upregulates ecdysone synthesis, which controls pupariation. Body size and pupariation are tightly coupled as a delay in the latter extends larval life and larvae reach pupariation with a larger body size. An intriguing result from these experiments was the observation that specific inactivation of *torso* in the prothoracic gland by means of RNAi downregulation caused a stronger delay in metamorphosis than the complete absence of the Torso receptor in *torso* null mutants. The discrepancy between the RNAi and the null phenotypes prompted the suggestion that Torso signalling might be required in other sites with an opposite effect [28]. Indeed, very recently Torso has been found to be present and have a role in the larval body in regulating body size in a manner opposite to that of Torso in the prothoracic gland. However, the effect of Torso on body size appears to be independent of any effect on the time of pupariation. This is explained by the proposal that the Torso pathway in the prothoracic gland regulates ecdysone synthesis, while in the fat body it controls insulin signalling [30].

Research into the regulation of ecdysone synthesis revealed an important finding with respect to Torso

signalling: in the prothoracic gland the Torso receptor is not triggered by Trunk but instead by another ligand belonging to the same family of proteins, namely the prothoracicotropic hormone (PTTH). Indeed, recently it has been solved at the structural level how PTTH binds to the Torso receptor [31]. PTTH is produced in some neurons that synapse within the prothoracic gland, and thus this hormone would directly activate the Torso receptor present in this gland [28]. However, PTTH produced by the same neurons is also secreted in the haemolymph, and it reaches two groups of neurons at the larval body wall. At this site, it also activates the Torso receptor, which is necessary for the light avoidance of *Drosophila* larvae. In particular, Torso signalling appears to impinge on the phototransduction pathway to facilitate the activation of a particular kind of cation channel. Interestingly, a single mechanism, Torso transduction, might thus regulate both developmental progression and innate behavioural decisions and optimize fitness and survival [29].

Which other elements acting on Torso activation in the embryo are involved in these other settings of Torso signalling? It is clear that Nasrat/Polehole/Closca proteins are not involved, while the participation of Tsl in these processes is still a matter of debate [32,33]. On the one hand, *tsl* is expressed in the prothoracic gland and is involved in regulating developmental timing as *tsl* mutant larvae have a delay in the onset of pupariation. Moreover, specific downregulation of *tsl* in this gland leads to a delay in pupariation, thereby supporting the notion that Tsl is involved [32]. On the other hand, some studies propose that Tsl acts independently of Torso in regulating developmental timing and body size [33]. Indeed, although both Torso and Tsl are involved in regulating developmental timing, they have opposite effects on body size regulation. While the downregulation of Torso in the prothoracic gland or ablation of PTTH-producing neurons results in larger adults due to prolonged feeding as larvae [28,34], *tsl* mutants or *torso;tsl* double mutants are smaller than wild-type larvae [33]. Moreover, *torso;tsl* double mutants show a dramatic delay in reaching pupariation, thereby indicating that *tsl* has an additive rather than epistatic effect on *torso* mutations [33].

## 8. Complexity from simplicity?

How can such a complex system of Torso receptor activation have been built for transducing positional information from the ovary into the embryo. A first indication that activation of the Torso pathway might have recruited some pre-existing elements came with the observation of *tsl* being expressed in sites where no Torso activity had been reported [11,35,36], thereby suggesting a Torso-independent function for Tsl. Similarly, Tsl homologues were also found to be present in animals where terminal morphogenesis does not rely on Torso signalling [37–39]. Indeed, the role of the Torso pathway in embryonic morphogenesis appears to be exclusive to highly evolved types of insect [40]. Thus, a picture emerges of an initially simpler mechanism of Torso activation in which the simple delivery of a ligand (whether cleaved or not) would activate the receptor in the absence of Tsl, as it might be the case for PTTH [33]. Such a basic scenario could have been recruited to embryonic patterning by co-opting specific eggshell proteins to ensure the transmission of Tsl from ovarian follicle cells to the early embryo, thus accounting for the temporal gap and the restricted spatial activation of the Torso receptor.

## 9. Two ligands, two functions?

As mentioned, the multiple use of the Torso pathway in distinct settings is coupled to the presence of two different ligands, Trunk for embryonic patterning and PTTH for regulating pupariation and light avoidance behaviour; whether Trunk or PTTH is the ligand of Torso in the fat body is not yet known. Trunk and PTTH are similar at the sequence level; they form a separate cluster among the cysteine-knot proteins [28,32] and each one is the closest paralogue of the other, probably because of a duplication of an ancestral gene at the base of holometabolous insects [32]. Interestingly, Trunk and PTTH display the same regional specificity in other insects: Trunk in the oocyte and PTTH in brain neurons [32]. However, the diversity of ligands is not a result of each being functionally organ- or tissue-specific as, at least in *Drosophila*, each ligand, if appropriately expressed, can activate Torso in all settings. Thus, forced expression of PTTH in the blastoderm can activate the Torso receptor to generate embryonic terminal structures [28,33] and forced ubiquitous expression of an active form of Trunk can activate the Torso receptor to advance pupariation [32].

Gene duplication is a first step in one of the mechanisms to extend a given gene function to new tissues or organs. In other cases, the same result is obtained by the acquisition of new enhancers in the regulatory region of a given gene, without the need for gene duplication. However, a classical assumption posits that gene duplication allows for the acquisition of

---

**Box 1.** Outstanding questions.

— Where and how is Trunk cleaved?
— Which is the active form of Trunk?
— Is Trunk secretion linked to cleavage?
— How is Tsl and membrane pore formation coupled to Trunk activity?
— Does Tsl perform a similar function in its Torso-dependent and Torso-independent role(s)?

---

diversity without modification of the features of the original gene (e.g. [41]). In the light of this framework, it is particularly appropriate to address whether PTTH and Trunk might each have a distinct capacity to trigger Torso activation. In fact, at least in SR2+ cells, PTTH is a weaker activator of the Torso pathway than Trunk, whether due to a different affinity for the Torso receptor or to differences in generating an active form of the ligand [27]. Consistent with the latter, and physiologically more relevant, while both proteins can activate the Torso receptor in the *Drosophila* early embryo, PTTH can perform this function in the absence of Tsl function. By contrast, Trunk is fully dependent on Tsl [33]. However, it is not clear whether these differences can be attributed to specific features of the proteins or to cell context specificities. Thus, while general expression of the full-length Trunk protein can activate the Torso pathway in *Drosophila* S2R+ cells [27], it does not do so in embryonic patterning or in pupariation [14,32]. Only a truncated form of the Trunk protein has some effect when expressed in the latter settings [12,32]. Indeed, while Trunk and PTTH are the closest paralogues and share sequence similarities such as the cysteine residues that contribute to the cysteine-knot motif, they differ in other cysteine residues that appear to be critical for the protein structure and/or protein interactions [9,34,42]. It remains to be established whether these or other differences might account for specific features of each protein that lead to distinct mechanisms to elicit Torso receptor activation.

Many new issues have emerged since we last reviewed Torso RTK signalling, and many questions remain to be answered (see box 1). Now, as then, the Torso system is a powerful tool through which to identify a variety of mechanisms that are likely to operate in many RTK pathways. Hopefully, this review will contribute to broadening the impact of such advances on the variety of biological phenomena that rely on cell communication and, as in the past with our previous review, might bring new researchers and approaches to this field. The challenge goes on.

Data accessibility. This article has no additional data.
Competing interests. We declare we have no competing interests.
Funding. Work in our laboratory is supported by the Generalitat de Catalunya and the Spanish Ministerio de Ciencia e Innovacion.

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
