## [Reviewer comments · Open Biology]

Review History

RSOB-18-0180.R0 (Original submission)

Review form: Reviewer 1

Recommendation

Accept with minor revision (please list in comments)

Are each of the following suitable for general readers?

- a) **Title**
Yes
- b) **Summary**
Yes
- c) **Introduction**
Yes

Is the length of the paper justified?

Yes

Should the paper be seen by a specialist statistical reviewer?

No

Is it clear how to make all supporting data available?

Not Applicable

Is the supplementary material necessary; and if so is it adequate and clear?

Not Applicable

Do you have any ethical concerns with this paper?

No

Comments to the Author

This is an excellent review of the input layer of the Torso signaling pathway, written by the leading experts in this system. The authors discuss recent progress in understanding the processes of active ligand generation, bringing together new connections with the membrane attack pore forming complexes, experiments with the Torso/Trunk/Tsl like system in cultured cells, and studies of postembryonic effects of Torso activation by PTH. The review is written in a very thoughtful way and makes an interesting read for anyone interested in problems at the intersections of biochemistry, cell biology, development. I suggest that the authors cite a recent review by Goyal et al (Developmental Biology, 2018), which focuses on the processes downstream of active Torso and a structural study by Daryl Klen (Molecular Cell, 2015). Otherwise the paper is good to go. I would include a better schematic for postembryonic signaling.

Review form: Reviewer 2

Recommendation

Major revision is needed (please make suggestions in comments)

Are each of the following suitable for general readers?

- a) **Title**
Yes
- b) **Summary**
Yes
- c) **Introduction**
Yes

Is the length of the paper justified?

Yes

Should the paper be seen by a specialist statistical reviewer?

No

Is it clear how to make all supporting data available?

Not Applicable

Is the supplementary material necessary; and if so is it adequate and clear?

Not Applicable

Do you have any ethical concerns with this paper?

No

Comments to the Author

This review article by Mineo et al. aims to cover the different contexts in which the Torso receptor tyrosine kinase has been shown to play a role in *Drosophila*, and how it might be activated in these contexts. The senior authors group has made seminal contributions to the study of the best known role of Torso, where it is locally activated at the embryo termini and controls terminal patterning.

Mineo et al first cover what is known about how Torso activity is restricted to the termini, including some quite recent studies from themselves and others. This coverage is comprehensive and care is taken to acknowledge all studies and their strengths and weaknesses. I would suggest, however, that the quite low penetrance of the rescue phenotypes observed in the Mineo et al 2018 Genetics paper should be mentioned to ensure that for this study this caveat is mentioned (as they have done for other studies).

The sections on other roles of Torso and how it is activated, and why there might be a need for a second ligand, are not as well written. The study by Jun et al (2016) showing a role for Tor in the fat body is mentioned, but what ligand is used in this context is not mentioned (even if not known, this should be stated.). Overall I was not left with a clear sense of what the authors think regarding why there are two different ligands for Tor and why Trk is used in one context and Pth in another, and why different mechanisms for controlling the activity of these ligands might exist. The final two sections ("complexity from simplicity" and "two ligands, two functions") are aimed at addressing these questions, but do not give a clear picture. On p10 line 5 they say that a picture emerges of an initially simpler mechanism where a ligand can always activate the receptor. However no evidence for the existence of this is presented. The review is also missing a discussion of the way forward and what remains to be discovered. Clarity on these questions would markedly improve this review and ensure it is useful for a non specialist.

Throughout the manuscript there are numerous grammatical issues in need of attention, and some sentences/sections with missing references. These all need addressing. Also, nowhere at the start does it state this is all in *Drosophila*!

Decision letter (RSOB-18-0180.R0)

Dear Dr Casanova

We are pleased to inform you that your manuscript RSOB-18-0180 entitled "The trigger (and the restriction) of Torso RTK activation" has been accepted by the Editor for publication in Open Biology. The reviewer(s) have recommended publication, but also suggest some minor revisions to your manuscript. Therefore, we invite you to respond to the reviewer(s)' comments and revise your manuscript.

Please submit the revised version of your manuscript within 14 days. If you do not think you will be able to meet this date please let us know immediately and we can extend this deadline for you.

- 1) A text file of the manuscript (doc, txt, rtf or tex), including the references, tables (including captions) and figure captions. Please remove any tracked changes from the text before submission. PDF files are not an accepted format for the "Main Document".
- 2) A separate electronic file of each figure (tiff, EPS or print-quality PDF preferred). The format should be produced directly from original creation package, or original software format. Please note that PowerPoint files are not accepted.
- 3) Electronic supplementary material: this should be contained in a separate file from the main text and meet our ESM criteria (see <http://royalsocietypublishing.org/instructions-authors#question5>). All supplementary materials accompanying an accepted article will be treated as in their final form. They will be published alongside the paper on the journal website and posted on the online figshare repository. Files on figshare will be made available approximately one week before the accompanying article so that the supplementary material can be attributed a unique DOI.

Online supplementary material will also carry the title and description provided during submission, so please ensure these are accurate and informative. Note that the Royal Society will not edit or typeset supplementary material and it will be hosted as provided. Please ensure that the supplementary material includes the paper details (authors, title, journal name, article DOI). Your article DOI will be 10.1098/rsob.2016[last 4 digits of e.g. 10.1098/rsob.20160049].

- 4) A media summary: a short non-technical summary (up to 100 words) of the key findings/importance of your manuscript. Please try to write in simple English, avoid jargon, explain the importance of the topic, outline the main implications and describe why this topic is newsworthy.

Images

Data-Sharing

It is a condition of publication that data supporting your paper are made available. Data should be made available either in the electronic supplementary material or through an appropriate

repository. Details of how to access data should be included in your paper. Please see <http://royalsocietypublishing.org/site/authors/policy.xhtml#question6> for more details.

Data accessibility section

Sincerely,

The Open Biology Team
<mailto:openbiology@royalsociety.org>

Reviewer(s)' Comments to Author:

Referee: 1

Comments to the Author(s)

This is an excellent review of the input layer of the Torso signaling pathway, written by the leading experts in this system. The authors discuss recent progress in understanding the processes of active ligand generation, bringing together new connections with the membrane attack pore forming complexes, experiments with the Torso/Trunk/Tsl like system in cultured cells, and studies of postembryonic effects of Torso activation by PTTH. The review is written in a very thoughtful way and makes an interesting read for anyone interested in problems at the intersections of biochemistry, cell biology, development. I suggest that the authors cite a recent review by Goyal et al (*Developmental Biology*, 2018), which focuses on the processes downstream of active Torso and a structural study by Daryl Klen (*Molecular Cell*, 2015). Otherwise the paper is good to go. I would include a better schematic for postembryonic signaling.

Referee: 2

Comments to the Author(s)

This review article by Mineo et al. aims to cover the different contexts in which the Torso receptor tyrosine kinase has been shown to play a role in *Drosophila*, and how it might be activated in these contexts. The senior authors group has made seminal contributions to the study of the best known role of Torso, where it is locally activated at the embryo termini and controls terminal patterning.

Mineo et al first cover what is known about how Torso activity is restricted to the termini, including some quite recent studies from themselves and others. This coverage is comprehensive and care is taken to acknowledge all studies and their strengths and weaknesses. I would suggest, however, that the quite low penetrance of the rescue phenotypes observed in the Mineo et al 2018

Genetics paper should be mentioned to ensure that for this study this caveat is mentioned (as they have done for other studies).

The sections on other roles of Torso and how it is activated, and why there might be a need for a second ligand, are not as well written. The study by Jun et al (2016) showing a role for Tor in the fat body is mentioned, but what ligand is used in this context is not mentioned (even if not known, this should be stated.). Overall I was not left with a clear sense of what the authors think regarding why there are two different ligands for Tor and why Trk is used in one context and Pth in another, and why different mechanisms for controlling the activity of these ligands might exist. The final two sections ("complexity from simplicity" and "two ligands, two functions") are aimed at addressing these questions, but do not give a clear picture. On p10 line 5 they say that a picture emerges of an initially simpler mechanism where a ligand can always activate the receptor. However no evidence for the existence of this is presented. The review is also missing a discussion of the way forward and what remains to be discovered. Clarity on these questions would markedly improve this review and ensure it is useful for a non specialist.

Throughout the manuscript there are numerous grammatical issues in need of attention, and some sentences/sections with missing references. These all need addressing. Also, nowhere at the start does it state this is all in *Drosophila*!

Author's Response to Decision Letter for (RSOB-18-0180.R0)

See Appendix A.

Decision letter (RSOB-18-0180.R1)

08-Nov-2018

Dear Dr Casanova

We are pleased to inform you that your manuscript entitled "The trigger (and the restriction) of Torso RTK activation" has been accepted by the Editor for publication in Open Biology.

Sincerely,

The Open Biology Team
mailto: openbiology@royalsociety.org

Appendix A

Referee: 1

This is an excellent review of the input layer of the Torso signaling pathway, written by the leading experts in this system. The authors discuss recent progress in understanding the processes of active ligand generation, bringing together new connections with the membrane attack pore forming complexes, experiments with the Torso/Trunk/Tsl like system in cultured cells, and studies of postembryonic effects of Torso activation by PTH. The review is written in a very thoughtful way and makes an interesting read for anyone interested in problems at the intersections of biochemistry, cell biology, development. I suggest that the authors cite a recent review by Goyal et al (Developmental Biology, 2018), which focuses on the processes downstream of active Torso and a structural study by Daryl Klen (Molecular Cell, 2015). Otherwise the paper is good to go. I would include a better schematic for postembryonic signaling.

We are pleased by the positive evaluation of the referee. We cite in the final version the papers he/she suggests and have added in figure 2 a simpler "arrow scheme" to complement the "morphological scheme"

Referee: 2

This review article by Mineo et al. aims to cover the different contexts in which the Torso receptor tyrosine kinase has been shown to play a role in Drosophila, and how it might be activated in these contexts. The senior authors group has made seminal contributions to the study of the best known role of Torso, where it is locally activated at the embryo termini and controls terminal patterning.

Mineo et al first cover what is known about how Torso activity is restricted to the termini, including some quite recent studies from themselves and others. This coverage is comprehensive and care is taken to acknowledge all studies and their strengths and weaknesses. I would suggest, however, that the quite low penetrance of the rescue phenotypes observed in the Mineo et al 2018 Genetics paper should be mentioned to ensure that for this study this caveat is mentioned (as they have done for other studies).

We are also pleased by the overall positive appreciation of the referee and have followed his/her suggestions. In this regard we mention the quite low penetrance of the phenotypes observed in the Mineo et al 2018 Genetics paper.

The sections on other roles of Torso and how it is activated, and why there might be a need for a second ligand, are not as well written. The study by Jun et al (2016) showing a role for Tor in the fat body is mentioned, but what ligand is used in this context is not mentioned (even if not known, this should be stated.).

We did mention that it is not known what ligand is used in the fat body in the legend to figure 2 in the original manuscript. We now also mention it in the main text in the new version.

Overall I was not left with a clear sense of what the authors think regarding why there are two different ligands for Tor and why Trk is used in one context and Pttb in another, and why different mechanisms for controlling the activity of these ligands might exist. The final two sections (“complexity from simplicity” and “two ligands, two functions”) are aimed at addressing these questions, but do not give a clear picture.

We agree with the referee. We think we can not answer this why question (if such an answer does exist!). We comment what might account for this diversity although, in our opinion, there is no clear explanation accounting for this fact. This is probably the reason not to be able to give a clear picture: we present and discuss the known facts but in our view there is probably not a definitive conclusion.

On p10 line 5 they say that a picture emerges of an initially simpler mechanism where a ligand can always activate the receptor. However no evidence for the existence of this is presented.

We have modified the sentence along the line suggested by the referee.

The review is also missing a discussion of the way forward and what remains to be discovered. Clarity on these questions would markedly improve this review and ensure it is useful for a non specialist.

Our manuscript included already a specific box on Outstanding questions. We now mention this box in the main text.

Throughout the manuscript there are numerous grammatical issues in need of attention, and some sentences/sections with missing references. These all need addressing.

The person in charge of proofreading at the Institute, who is an English professional, reviewed the initial manuscript. We have also added some references but we think that the comment from the reviewer might be motivated because, in the case that the content of the different sentences in a section might be attributed to the same set of original papers, we chose to include those references at the end of the section instead of after each sentence.

Also, nowhere at the start does it state this is all in Drosophila!

We now state in the first page that the Torso pathway is a pathway in Drosophila.